# Biomaterials in Traumatic Brain Injury: Perspectives and Challenges

**DOI:** 10.3390/biology13010021

**Published:** 2023-12-29

**Authors:** Sarah Aqel, Najlaa Al-Thani, Mohammad Z. Haider, Samar Abdelhady, Asmaa A. Al Thani, Firas Kobeissy, Abdullah A. Shaito

**Affiliations:** 1Medical Research Center, Hamad Medical Corporation, Doha P.O. Box 3050, Qatar; 2Research and Development Department, Barzan Holdings, Doha P.O. Box 7178, Qatar; 3Department of Basic Medical Sciences, College of Medicine, QU Health, Qatar University, Doha P.O. Box 2713, Qatar; mh1704315@student.qu.edu.qa; 4Faculty of Medicine, Alexandria University, Alexandria 21544, Egypt; samar.abdelhady606@gmail.com; 5Biomedical Research Center and Department of Biomedical Sciences, College of Health Science, QU Health, Qatar University, Doha P.O. Box 2713, Qatar; aaja@qu.edu.qa; 6Department of Neurobiology, Center for Neurotrauma, Multiomics & Biomarkers (CNMB), Morehouse School of Medicine, 720 Westview Dr. SW, Atlanta, GA 30310, USA; 7Biomedical Research Center, Department of Biomedical Sciences at College of Health Sciences, College of Medicine, Qatar University, Doha P.O. Box 2713, Qatar

**Keywords:** traumatic brain injury, TBI, biomaterials, hydrogels, self-assembling peptides, electrospinning

## Abstract

**Simple Summary:**

There are no Food and Drug Administration (FDA)-approved drugs for traumatic brain injury (TBI). The available treatments have limitations, including limited access to the injury site, mainly due to the complex pathology of TBI and the presence of the blood–brain barrier. This review collects and discusses the available literature on the use of biomaterials, mainly hydrogels, including self-assembling peptides and electrospun nanofibers to enhance the therapeutic outcomes of TBI. The challenges and limitations that such an approach faces are also exposed.

**Abstract:**

Traumatic brain injury (TBI) is a leading cause of mortality and long-term impairment globally. TBI has a dynamic pathology, encompassing a variety of metabolic and molecular events that occur in two phases: primary and secondary. A forceful external blow to the brain initiates the primary phase, followed by a secondary phase that involves the release of calcium ions (Ca^2+^) and the initiation of a cascade of inflammatory processes, including mitochondrial dysfunction, a rise in oxidative stress, activation of glial cells, and damage to the blood–brain barrier (BBB), resulting in paracellular leakage. Currently, there are no FDA-approved drugs for TBI, but existing approaches rely on delivering micro- and macromolecular treatments, which are constrained by the BBB, poor retention, off-target toxicity, and the complex pathology of TBI. Therefore, there is a demand for innovative and alternative therapeutics with effective delivery tactics for the diagnosis and treatment of TBI. Tissue engineering, which includes the use of biomaterials, is one such alternative approach. Biomaterials, such as hydrogels, including self-assembling peptides and electrospun nanofibers, can be used alone or in combination with neuronal stem cells to induce neurite outgrowth, the differentiation of human neural stem cells, and nerve gap bridging in TBI. This review examines the inclusion of biomaterials as potential treatments for TBI, including their types, synthesis, and mechanisms of action. This review also discusses the challenges faced by the use of biomaterials in TBI, including the development of biodegradable, biocompatible, and mechanically flexible biomaterials and, if combined with stem cells, the survival rate of the transplanted stem cells. A better understanding of the mechanisms and drawbacks of these novel therapeutic approaches will help to guide the design of future TBI therapies.

## 1. Introduction

Globally, traumatic brain injury (TBI) is one of the prime causes of mortality and long-term physical and cognitive disabilities. The global incidence rate of TBI is estimated to be 939 per 10,000 individuals, and these are most often caused by falls, motor vehicle accidents, wars, and sports [1,2,3,4,5]. TBI is more frequent in low- and middle-income countries, severely affecting young people and adolescents [1,2,3,4,6]. TBI can have a considerable impact on a patient’s quality of life, and its consequences persist for a prolonged period after the injury. TBI survivors often face significant socioeconomic consequences, such as job loss and divorce, further contributing to the economic burden of TBI [7,8]. Furthermore, epilepsy, sleep difficulties, neurodegenerative illnesses, neuroendocrine dysregulation, and psychological issues are all subsequent pathological symptoms caused by a single TBI or repeated insults to the brain [9,10,11,12,13]. In a study of American army soldiers who reported post-TBI symptoms, more than 17% of participants had a positive diagnosis of post-traumatic stress syndrome [14]. Notably, TBI may precipitate Alzheimer’s disease (AD), Parkinson’s disease (PD), or total cognitive impairment [15,16].

TBI severity is determined by the primary injury, which is the most important prognostic factor [17]. Depending on the degree of neurological damage, TBI is classified as mild, moderate, or severe. At least 75% of all TBIs recorded in the United States are categorized as minor or mild concussions, covering the spectrum of mild TBI (mTBI) [18]. While mTBI patients usually recover on their own within days to months, it is important to note that a significant percentage of patients, ranging from 30% to 53%, may continue to experience disabling symptoms for at least a year following brain injury [19,20]. The complex pathology of TBI develops at the time of mechanical impact and the initial injury to the brain (TBI primary phase). TBI continues to evolve over time throughout its secondary phase. The primary phase is immediately triggered by an external mechanical insult such as acceleration, deceleration, or rotational forces, while the secondary phase may occur minutes or days after the primary injury [21,22]. Primary injury usually manifests as elevated intracranial pressure, nerve damage, vascular damage, tissue swelling, and hypoxic damage [23]. The secondary brain injury that follows is triggered by a variety of molecular and cellular events, which include oxidative stress, neuronal excitotoxicity, mitochondrial dysfunction, inflammation, edema, and neuronal cell death, leading to further cerebral damage [21,22,24,25,26]. Figure 1 provides a synopsis of the main events of TBI.

In response to a physical force, the white matter of the brain deforms, leading to diffuse axonal damage and release of calcium ions (Ca^2+^) from intracellular stores [27,28]. After an excessive release of excitatory neurotransmitters such as glutamate, post-synaptic terminals become depolarized due to an influx of Ca^2+^, thereby resulting in hypermetabolism, which eventually leads to metabolic depression lasting several days [29,30]. High Ca^2+^ levels disrupt several intracellular functions, including the generation of a state of cellular hypoxia. Under hypoxia, the brain is compelled to switch to glycolytic metabolism, which leads to the accumulation of lactic acid [20,31]. Mitochondria play crucial roles in TBI pathology. Increased Ca^2+^ concentration induces excess mitochondrial Ca^2+^ absorption, leading to mitochondrial membrane permeabilization, mitochondrial dysfunction, and an enhanced state of oxidative stress, illustrated by the generation of reactive oxygen species (ROS) [32]. Oxidative stress exacerbates other TBI-associated pathological pathways, such as cytoskeletal damage, via calpain activation, and neuroinflammation, via glial cell activation [33,34,35,36]. After injury to the brain, oxidative stress changes the crucial architecture of tight junction proteins at the blood–brain barrier (BBB), which is one of the most vital components of a healthy brain, acting as a barrier between the central nervous system (CNS) and the rest of the body [37]. Breaching of the BBB in TBI results in increased paracellular leakage [38]. The harsh microenvironment near the lesion site mainly drives the transformation of activated native neural stem cells (NSCs) into astrocytes. This process also results in the formation of glial fibrosis, which seals cavities between neurons [39]. Consequently, this tissue obstructs the transfer of electrical signals in functioning nerve cells at the affected site [40] and acts as a major physical barrier of axonal regeneration, which impedes recovery [41]. Astrocytes further promote BBB rupture after TBI by activating paracellular channels, physically disrupting astrocyte–endothelial junctions, and digesting BBB matrix proteins [42,43]. TBI-induced disruption of the BBB significantly contributes to TBI pathology, but may be exploited to pass therapeutics through the damaged BBB.

Despite gradual advances in TBI treatment, long-term damage following TBI remains a substantial healthcare concern. As of now, the Food and Drug Administration (FDA) has not approved any drug to treat TBI. The current standard of care for patients with moderate to severe TBI includes ventilation and oxygenation interventions, fluid management, hypothermic stimulation, intracranial pressure (ICP) control, cerebral perfusion pressure (CPP), blood pressure (BP) management, nutrition and glucose level management, and surgery [44]. Many of the available diagnostic and treatment options are limited by the exceedingly complicated pathology that follows brain injury. Importantly, the BBB, which primarily governs material access into the brain, inhibits the entry of micro- and macro-molecular therapies [45]. As a result, TBI therapy by the systemic or local delivery of medications is mostly ineffective [46,47,48]. Consequently, current therapeutic approaches are often constrained by two major obstacles: (1) ineffective delivery and retention, restricting therapeutic thresholds; and (2) off-target toxicity induced by treatments that target receptors of biochemical derangements rather than the derangements themselves, resulting in loss of function in off-target cells [49].

Recently, novel strategies have been developed to repair brain tissue damage. One approach that has gained significant attention as a potential TBI treatment is tissue engineering, which relies on the use of biomaterials alone or biomaterials in combination with stem cells and other factors (Figure 2). In relation to stem cell-based therapy, the low survival rate of transplanted stem cells is the most significant impediment to successful therapy [50], and biomaterials can enhance the survival of transplanted stem cells. Today, biocompatible three-dimensional biomaterials can be combined with cells and bioactive chemicals to repair tissue injury while preserving as much as possible of the anatomy of the injured tissues [51,52,53,54]. In this review, we examine the use of biomaterials as a potential therapeutic option for TBI. Specifically, this review discusses the promising uses of hydrogels, including self-assembling peptides, and electrospun fibers in TBI therapy. We discuss the reported applications of biomaterials in TBI, including when they are used alone as structural scaffolds or in combination with cells to support cell delivery and implantation, drugs to assist in drug delivery, growth, angiogenic, and adhesive factors, or extracellular matrix (ECM) proteins [50,51,52,53,54]. This review will not include a discussion of nanomedicine and nanoparticles, another approach under investigation for TBI therapy [55].

## 2. Biomaterials in Neurological Disorders

Biomaterials are being investigated for therapeutic effects in a range of neurological disorders, including TBI, spinal cord injury, AD, PD, and stroke [56]. In these investigations, biomaterials have been shown to exert therapeutic effects on their own or in combination with other factors, such as cells, growth, neurotrophic, or angiogenic factors, and extracellular matrix (ECM) components [50,56].

Brain tissue is the most delicate, soft, and elastic tissue of the human body [57,58]. In addition, the brain has heterogeneous cellular composition and stiffness. There is heterogeneity within the individual anatomical structures as well [59]. Therefore, biomaterials for use in the delicate brain should have a defined set of design principles. These design principles vary according to the type of brain disorder, but they share some commonalities. Biomaterials for brain therapy should be biocompatible with the sensitive neural tissue [58]. To achieve biocompatibility, material mechanical properties and those of the brain should match. A material stiffer than brain tissue will increase gliosis and aggravate outcomes. A biomaterial softer than the brain tissue will not be stable [58,60]. In addition, the biomaterial should have minimal swelling in order to prevent the squeezing of brain tissue in the confined space of the skull, thereby increasing intracranial pressure. Consequently, injectable and shape-adjusting biomaterials perform better than stiff biomaterials as they can fit into heterogeneous cavities and their application usually requires less invasive surgical operations. In addition, biomaterials for brain therapy must be biodegradable and resorbable [58]. Nonbiodegradable brain implants or those used in the long-term were shown to induce inflammation, scarring, and neuronal cell death [58,61]. Inflammation and immunogenicity are major limitations of biomaterials, but they may be tuned down by choosing a biomaterial with similar physical properties to brain tissue (i.e., low-elastic nature and low interfacial tension to minimize the adhesion of immune cells) [58]. Design principles will also depend on the proposed use of the biomaterial. For example, biomaterials used in cell-based therapeutics should promote cell adhesion and prevent cell aggregation. In addition, they should be compatible with live cells and biodegradable. On the other hand, for biomaterials used in drug delivery, the stability of the biomaterial, drug solubility, and extent of tissue penetration are the principles that should be considered [58].

Biomaterials can be obtained from either natural or synthetic sources. Natural biomaterials are often fabricated from either human and mammalian ECM or from other organisms. ECM-derived biomaterials include hyaluronic acid (HA), heparin, collagen, fibrin, laminin and other ECM peptides, and proteins. On the other hand, chitosan, silk, methylcellulose, alginate, and Matrigel™ are natural biomaterials obtained from other organisms [58]. Synthetic materials often employed for therapy of neurological disorders include polyethylene glycol (PEG), poly(d,l-lactic acid), polyglycolic acid (PGA), poly(d,l-lactic acid co-glycolic acid) (PLGA), poly(d-lysine), poly(sebacic acid) (PSA), and polycaprolactone (PCL) [62]. Natural biomaterials that are obtained from human or mammalian ECM have characteristics that match the ECM of the damaged tissue, and are therefore less immunogenic. They also have the right adhesion molecules required to adhere to the injured area [58]. Natural or biological biomaterials are well known for their high bioactivity, good biocompatibility and degradation properties, and resemblance to the ECM [63]. In comparison, synthetic biomaterials have the advantages of stability and easier tuning to the requirements of the desired use. For example, synthetic hydrogels can be manufactured under controlled conditions, which provides the ability to predict their mechanical and physical properties and behaviors [63,64]. Moreover, they are easily sterilized and less likely to produce an immune response. This level of control offers a notable advantage over natural biomaterials [65]. However, synthetic biomaterials suffer from low biocompatibility and a poor ability to induce tissue regeneration [63,64]. However, this may be overcome by empowering them with different kinds of functional molecules such as adhesive, angiogenic, or neurotrophic molecules [62]. Table 1 summarizes the main differences between biomaterials from natural or synthetic sources.

The biomaterials that have shown success in therapy of neurological disorders are mainly injectable hydrogels, electrospun fibers, and nano- and microparticles. Hydrogels are mainly made up of water and can form scaffolds of polymeric three-dimensional (3D) networks crosslinked by either chemical bonds or physical contact [66]. Hydrogels can be manufactured by crosslinking hydrophilic polymers, a process affected by physical factors (i.e., light and temperature) and chemical factors (i.e., pH and ionic concentration) [67,68]. For materials to be considered hydrogels, 10–20% of their total weight must consist of water [69], granting them flexibility [70]. Their highly hydrophilic nature allows them to transport various soluble molecules, making them valuable in biomedical contexts. They allow for the diffusion of nutrients, oxygen, drugs, and other factors needed to maintain endogenous or implanted cells [71]. Notably, hydrogels mirror essential physical traits of native tissues, since they encompass high water content, comparable elasticity ranges, and effective mass transfer mechanisms. Their porosity and ability to reshape their forms allow them to fill cavities sustained by disease or injury [72]. They may be modified to resemble the mechanical characteristics of the brain tissue to reduce immunogenicity and enhance therapeutic outcomes. Overall, synthetic hydrogels have better controllability, immunogenicity, and histocompatibility and are more amenable for tuning [8]. In fact, recent studies on therapy of neurological disorders have mainly used synthetic hydrogels [73,74].

Covalently crosslinked hydrogels and self-assembled hydrogels are the two main types of hydrogels that differ in their synthesis procedures [75,76]. Polymeric covalently crosslinked hydrogels are considered more stable to changes in environmental factors, such as temperature and pH, owing to their covalently linked monomers [77]. In addition, they are mostly less deformable, but stiffer, requiring surgery for their implant inside the human body [78,79]. They can be made of synthetic materials or materials from natural sources, such as hyaluronic acid, fibroin, chitosan, collagen, and alginate [80,81,82,83,84,85]. The main benefits of these natural molecules are being biodegradable, easy to acquire, highly biocompatible, and containing particular cell adhesion molecules [86]. Polysaccharides and glycosaminoglycans, some of which are components of the ECM such as HA, make up the majority of biologically generated hydrogels [87,88,89]. Collagen and HA are the natural polymers most commonly employed to generate the hydrogels used in brain tissue engineering [90,91]. Nevertheless, these natural hydrogel scaffolds lack homogeneity, due to variations between batches [87].

In contrast, synthetic hydrogels are often chemically stable, but have poor cell adhesion properties because they are biologically inert. However, they can also be modified and functionalized for use in neural tissue engineering. Nowadays, polyethylene glycol (PEG) is a main component of synthetic hydrogels that are applied in CNS therapy [92,93,94].

Biodegradable scaffolds can be synthesized from natural or synthetic materials. The natural materials that are often used include collagen, fibroin, chitosan, and HA [76,95,96,97]. On the other hand, poly Ɛ-caprolactone (PCL), poly L-lactic acid (PLA), and polyurethane are all examples of materials used to synthesize synthetic biodegradable scaffolds [98,99,100]. Yet, PCL is hydrophobic, resulting in a lack of cell interaction and poor cell adhesion and proliferation. As a solution to this problem, copolymer biodegradable scaffolds have been established by combining two or more chemical species into the polymer, converting the scaffold from hydrophobic to hydrophilic. The main two examples of copolymers are poly D, L-lactide-co-glycolic acid (PLGA) and poly Ɛ-caprolactone-co-ethyl ethylene phosphate (PCLEEP) [101,102].

Another frequently employed category of hydrogels is referred to as “smart” or stimuli-responsive hydrogels. Smart hydrogels have a high degree of sensitivity to even minor changes in their external surroundings such as temperature and pH. This adaptability enables them to promptly adjust their physical properties, including mechanical strength and swelling capacity, in response to these changes [7,8]. Stimuli-responsive hydrogels have a variety of subtypes, which include, but are not confined to, thermoresponsive, photoresponsive, electroresponsive, and bioresponsive (smart) hydrogels.

To make electrospun fibers, a viscoelastic polymer solution is uniaxially stretched to create a nanofibrous mesh as part of the electrospinning scaffolding process [103]. Compared with other biomaterials, electrospun nanofibers present distinct advantages which include their simple preparation, high loading capability, and adjustable mechanical properties [104,105]. Electrospun nanofibers mimic the hierarchical fibrillar arrangement of collagen, laminin, and other fibrils of the ECM. This resemblance is the basis of the interest in nanofibrous scaffolds for tissue engineering [106,107,108,109]. Electrospun nanofibers can guide axons [58,110]. Nevertheless, scaffolds of electrospun fibers allow for limited cell migration, but this can be enhanced by including the fibers in hydrogels [58]. The hydrophilicity or hydrophobicity of nanofibers can be carefully tailored to optimize their compatibility with the aqueous environment of the brain, influencing their effectiveness in therapeutic applications [111]. Electrospun nanofibers mimic other features of the cellular ECM, including a large surface-area-to-volume ratio, high porosity, and similar mechanical properties [108]. These similarities allow electrospun nanofibers to enhance drug-loading efficiency and provide a faster response to the drugs they deliver [112]. In addition, they offer promising avenues for the development of controlled drug delivery systems. An important illustration is the utilization of two types of PLGA fiber mats loaded with nimodipine (a neuroprotective drug). These fiber mats demonstrated a prolonged and controlled release of the drug for a period of 4–8 days and reduced oxidative stress-induced death of neuronal, Schwan, and astrocyte cells in vitro [113].

Additionally, electroactive scaffolds have recently been under intensive investigation as they may help in the communication between brain neurons. For example, in situ polymerization was used to cover PCL and poly-l-lactide nanofibrous scaffolds with polypyrrole (Ppy) to create conductive sheaths [114]. In addition, biomolecules like collagen can be attached to the surface of nanofibrous scaffolds to improve their properties, including the enhancement of cell survival and attachment [115]. In fact, various biocompatible materials, both natural and synthetic, have been employed in the fabrication of electrospun nanofibers for brain tissue repair, including collagen, chitosan, silk, fibronectin, fibrinogen, PLA, PCL, PLGA, Ppy, as well as their composites formed by combining them with each other or other materials [116].

## 3. Biomaterials and Their Mechanisms of Action in TBI

Compared to other organ systems, therapeutic strategies such as providing increased oxygen supply to damaged tissue and regulating temperature have been less effective and more difficult to apply to injured tissues of the brain [117], a possible consequence of the complexity of the CNS or the inhospitable environment around the site of injury. The treatment of TBI currently relies mainly on surgical methods, whereas pharmacological treatments are still under exploration [118]. Limited drug diffusion into the brain across the BBB remains a major hurdle in TBI treatment; hence, the efficiency of current treatment options is restricted, warranting the use of other medical strategies for TBI therapy. The use of biomaterials, either alone (Figure 2, left panel), to fill brain cavities and stimulate endogenous cell repair, or in combination with cell-based therapies (Figure 2, right panel), to assist exogenous cell transplantation, are good candidate strategies in this regard [117]. In fact, both approaches have been employed in TBI therapy. Biologically active biomaterials may promote recovery and repair by themselves and can also be used as delivery agents for factors that assist in tissue regeneration, such as cells, ECM proteins and growth factors, angiogenic factors, anti-inflammatory cytokines, and antioxidants [119,120,121] (Figure 2). Furthermore, using biomaterials to deliver cells in combination with neurotrophic growth factors or ECM proteins can properly guide the differentiation of stem cells and assist in tissue regeneration [121].

### 3.1. Biomaterials Utilized in TBI Therapy

Both natural and synthetic biomaterials have been investigated in therapy of TBI. In fact, the enhancement of neurite outgrowth, differentiation of human neural stem cells, and nerve gap bridging of damaged neurons or transplanted stem cells have all been successfully accomplished in models of TBI using both natural and synthetic biomaterials [50,51,52,53,54]. Furthermore, hydrogels, including self-assembling peptides, and electrospun fibers have shown the most promise in TBI treatment, as will be detailed.

TBI creates tissue discontinuity along with disruptions in the organized structure of the nerve tracts, resulting in detrimental effects on brain function. Restoring the neural network is vital for optimal brain function, which can only be achieved by restoring the continuity of functional tissue at the cavity/injury site. Thus, TBI creates irregularly shaped cavities, requiring a biomaterial scaffold that can conform to the shape of the injury while maintaining a suitable environment for cellular integration and repair. The challenge faced in the restoration process lies in ensuring that a scaffold closely resembles the native tissue in terms of both biological composition and architectural characteristics [122,123,124]. Scaffolds for use in TBI may be tailored in an injury/tissue-specific manner by altering the 3D architecture of the scaffold, porous spaces within the scaffold, and its physiochemical characteristics such as stiffness and cell-binding ability. Hydrogels that satisfy several of these requirements have shown the most promise in TBI therapy thus far [125,126,127,128,129].

When used in combination with stem cell therapeutics, biomedically active scaffolds can aid in the assembly from single cells into tissues [130], providing a suitable environment for cellular regrowth and repair. Several studies have shown that using a scaffold supports better regeneration of transplanted stem cells than the transplantation of a cell suspension alone [131,132,133,134]. In this regard, scaffolds derived from biological sources have been shown to be more effective in terms of cell survival and behavioral recovery than scaffolds without biological materials [135]. Some natural biological scaffolds that can be used in these applications include collagen, gelatin, fibrin, and HA, whereas options for using synthetic scaffolds exist and include linear aliphatic polyesters, poly-anhydrides, and poly-orthoesters [64]. For brain tissue engineering and TBI recovery, the scaffold surface must be tuned to sustain neurogenesis of endogenous or implanted cells and enable guided axonal growth. Because of the trade-offs between bulk and surface qualities, scaffolds may need to be optimized using techniques like adding biomolecules and applying surface treatments for better biorecognition [136,137].

Recent advancements in biotechnology, particularly 3D bioprinting, have enabled precise control over the internal microstructures of scaffolds, including the manipulation of pore size and microchannels. This level of control allows for the creation of complex scaffold architectures that closely resemble the intricate structures found in native tissues [138]. Scaffolds containing crucial ECM components can mimic the natural microenvironment required for tissue regeneration, facilitating endogenous tissue repair processes [139]. Biomaterial properties, such as surface chemistry, topography, and matrix stiffness, can significantly influence cellular functionality and behavior, like cell differentiation, proliferation, and adhesion [140]. Therefore, it is essential to consider specific material parameters that impact cell behavior by effectively interacting with cells to promote tissue regeneration when developing synthetic biomaterials for the regeneration of functional tissues. These parameters include the physical and mechanical features of the material, its chemical composition, coating of biomaterials with native ECM macromolecules, functionalization of biomaterials with adhesion proteins, reduction of inflammatory responses typically associated with biomaterial implantation, loading of biomaterials with growth factors, and incorporation of pharmaceuticals [141]. As such, altering or choosing a biomaterial with specific physical/chemical properties can alter cellular behavior. For example, cell proliferation is supported by positively charged biomaterials, whereas cell differentiation can be promoted by a negatively charged biomaterial [142]. The ability to modify the characteristics of biomaterials allows for a high flexibility in their applications. A variety of scaffolds with potential applications in brain tissue repair exist. Hydrogels, including self-assembling peptides, and electrospun nanofiber scaffolds offer such flexibility of use and are the most promising biomaterials in TBI therapy. Nevertheless, each of these scaffolds is fabricated using a different set of methods, and hence each has unique properties and applications. Table 2 presents findings from studies that have tested the ability of biomaterials to support neuronal growth and aid in the regeneration of damaged brain tissues following TBI in experimental models of TBI. In general, these studies have yielded promising results, with various scaffolds showing the potential to improve outcomes in models of TBI.

Considering the number of factors that influence scaffold efficiency, a perfect scaffold for CNS tissue repair has not yet been developed, and research is still being conducted to optimize the compatibility of existing scaffolds and maximize the differentiation of transplanted or endogenous neural stem cells. The following sections collect studies that have used biomaterials for TBI therapy.

#### 3.1.1. Hydrogels

The hydrogel scaffold approach has shown the most promise thus far in TBI therapy because it does not exhibit mechanical/spatial restrictions, compared to synthetic polymer scaffolds, which must be molded into a specific shape before implantation. Solid systems such as nanofibers lack the fluidity present in hydrogels and therefore cannot fill irregularly shaped lesions as effectively [125,126,127,128,129]. In addition, the mechanical properties of hydrogels can be modified to mimic those of soft tissues, such as the brain, which facilitates the transmission of mechanical signals to cells, similarly to what occurs in natural tissues [164]. This is especially important for TBI recovery, in which the presence of the glial scar impedes neuronal signal transmission, neurogenesis, and functional recovery. The presence of a conductive hydrogel may mitigate these defects [41,165].

Chen et al. classified the uses of hydrogels in TBI therapy into the following: (1) hydrogels alone; (2) hydrogels as drug delivery tools; and (3) hydrogels as a cell therapy delivery tools [74]. For the repair of brain injuries, a number of synthetic hydrogels have been employed (Table 2), including poly(N-2-(hydroxypropyl)methacrylamide) (pHPMA) [166,167], poly(hydroxyethylmethacrylate) (pHEMA), and PEG [168]. pHPMA and pHEMA hydrogels must be implanted premade, requiring invasive surgery [169]. Several of these hydrogels have produced encouraging outcomes. For example, a macroporous pHPMA was created by heterophase separation, employing radical polymerization in a pore-forming solvent with a divinyl crosslinking agent to bridge the brain lesion. This scaffold also allowed for cell penetration, angiogenesis, axon growth, and ECM production [166,167,168].

A hybrid thermosensitive injectable hydrogel, FPGEGa, containing the antioxidant poly citrate-gallic acid (Ga), was combined with exosomes derived from stem cells of human exfoliated deciduous teeth and tested in a rat model of TBI. The hydrogel loaded with the exosomes decreased oxidative stress levels and ROS generation, reduced microglia-mediated inflammation, and helped restore motor function in TBI rats [170]. A self-healing semi-penetrating hydrogel of HA and chitosan enhanced the growth and differentiation parameters of encapsulated neural stem cells in vitro. Interestingly, the hydrogel had better biocompatibility, repair of brain injury, and functional recovery in zebrafish and rat models of TBI than a chitosan-based hydrogel. The hydrogel was proposed to act by providing an adaptable environment for cell spreading and migration [171]. An injectable tyramine-modified HA hydrogel (HT), prepared by dual enzyme crosslinking of the polymer, was loaded with nerve growth factor (NGF) and hBMMSCs mesenchymal stem cells. The hydrogel enhanced the survival and proliferation of neural cells through the release of neurotrophic factors and the regulation of neuroinflammation in a mouse model of TBI. Consequently, there was a recovery of neurological functions and an acceleration of the repair process post TBI [172].

Yao et al. employed a “smart” temperature-responsive hydrogel in a rat model of TBI. They developed a temperature-sensitive chitosan–cellulose hyaluronic acid/β-glycerophosphate (CS-HEC-HA/GP) hydrogel, transitioning from a liquid state below 25 °C to a hydrogel form at 37 °C. Furthermore, this CS-HEC-HA/GP hydrogel, when laden with human hUC-MSCs, improved the survival rate and retention of the enclosed hUC-MSCs. Additionally, it was able to stimulate neurogenesis and suppress cell apoptosis, leading to expedited brain structure reformation and neurological function restoration in TBI rats [173].

#### Self-Assembling Peptides

Self-assembling peptides (SAPs) form hydrogels in water and are the main type of self-assembled hydrogels. SAPs are composed of repeating units of amino acids that can form double-β-sheet structures in water [174] (Figure 3). They may possess hydrophilic or hydrophobic properties depending on the sequence of the amino acids that they are made of. For example, hydroxyl-containing amino acids make them more water-soluble [175,176]. Self-assembled scaffolds are characterized by soft and deformable structures due to the internal noncovalent forces that link their monomers, and as a result can take the shape of the damaged tissue when injected into damaged tissues or organs [177,178]. SAPs can be modified by the addition of a functional motif to the SAP peptide. The functional motif can have angiogenic, growth-promoting, neurotrophic, or ECM-adhesive properties, among others (Figure 3). For example, laminin moieties, which are important proteins present in the ECM of brain tissue, have been added to the SAP peptide [179].

In addition, SAPs have the capacity to form nanofibrous particles which may later facilitate hydrogel formation under the appropriate conditions, for example in the pH of the injured brain [176,180]. These self-assembling peptide nanofiber scaffolds (SAPNs) can be created from different oligopeptides or amphiphilic compounds that naturally aggregate to form nanofibers. These in turn form a fibrillar network under physiological ionic conditions [181]. SAPNs are known to have tissue-like water content, high porosity, and increased cell signaling from bioactive peptides that are presented in high density at the damaged site [181].

Several SAPs and SAPNs have been investigated in TBI therapy (Table 2). Guo et al. investigated the application of SAPNs in TBI by implanting their scaffolds into the brains of rats that had experienced acute brain injury. The results demonstrated a notable enhancement in neurological functions and accelerated tissue regeneration when compared to the control group. Furthermore, SAPNs virtually abolished cavitation in brain lesions, with lower levels of macrophages and astrocytes at the lesion site than in controls with secondary tissue loss [182]. SAPNs are also permissive to axonal growth, so that neural tracts can be partially restored and functional recovery is accomplished after brain injury [183,184,185]. Relatedly, a self-assembling peptide comprising a 14-amino-acid portion of ependymin, a neuroprotective extracellular glycoprotein that is capable of forming nanofibrous matrices, has demonstrated its ability to promote neuronal survival both in vitro and in vivo. This peptide was tested in an acute fluid percussion injury model of TBI in rats, yielding favorable outcomes [186].

RADA16-I, one of the most studied SAPNs, is made up of repeating sequences of alanine, lysine, and glutamate in the form of RADA. Its nanofibers have a diameter of approximately 10 nm and are highly hydrated. RADA16-I has been extensively researched for tissue culture applications and is available in liquid and powder form [187,188]. More importantly, it has also been shown to promote regeneration in experimental spinal cord and brain injuries (Figure 3) [189]. In a study conducted by Cheng et al., the self-assembling peptide RADA16 was combined with a laminin-derived IKVAV motif to create a nanofibrous hydrogel that had mechanical properties similar to brain tissue. This IKVAV sequence directed neural stem cells towards neuronal differentiation in laboratory tests, and in animal studies, the peptide hydrogel increased the survival of stem cells and decreased the formation of glial astrocytes. As a result, there was an improvement in brain tissue regeneration after six weeks [179]. In another study, Ohno et al. developed an amphiphilic peptide [(RADA)3-(RADG)] (mRADA)-tagged N-cadherin extracellular domain (Ncad-mRADA), which could be retained in mRADA hydrogels and injected into deep brain tissue to aid neuroblast migration. Administering Ncad-mRADA in neonatal cortical brain injuries effectively enhanced neuronal regeneration and functional recovery [190].

Another recent study demonstrated that transplantation of hMgSCs seeded in R-GSIK scaffold improved functional recovery following TBI. The presence of R-GSIK scaffold significantly increased the number of hMgSCs in the brain compared to control groups. Furthermore, hMgSCs seeded in R-GSIK effectively reduced injury volume, reactive gliosis, and apoptosis. Notably, hMgSCs seeded in R-GSIK exhibited a significant inhibition of Toll-like receptor 4 expression, as well as downstream signaling molecules including interleukin-1β and tumor necrosis factor. These findings suggest that the SAPN hMgSCs with R-GSIK may enhance brain injury healing by enhancing the implantation of neuronal stem cells and suppressing inflammation [154]. In the Fmoc-DIKVAV SAP scaffold, an anti-inflammatory and antiproliferative polysaccharide, fucoidan, was encapsulated to monitor neuronal cell inflammation in vitro and in vivo. The anti-inflammatory action of this scaffold maintained the growth, cytoskeletal reorganization, and trophic responses of brain neurons for an extended duration in culture and in a mouse model of TBI [191].

#### 3.1.2. Electrospun Nanofibers

Electrospun nanofibers have also been investigated in TBI therapy (Table 2). The effects of an L-lactide-caprolactone copolymer nanofiber net dressing were examined in a study by Sulejczak et al., revealing its ability to delay and alleviate damaging processes such as neurodegeneration, systemic inflammatory cell infiltration, and excessive formation of glial scars [150]. The anti-inflammatory properties of electrospun nanofiber scaffolds are particularly noteworthy. A recent study covalently bonded galactose to the surface of PCL nanofiber scaffolds, generating a polymer referred to as poly(L-lysine)-lactic acid (PLL-LBA). When implanted into a mouse model of TBI, this scaffold led to an increased survival of neurons 21 days post implantation, emphasizing its potential therapeutic impact. Furthermore, this scaffold demonstrated the capability of galactose to sustain an attenuated inflammatory response when compared to control PCL nanofibers devoid of galactose [192]. Uniaxially aligned electrospun PCL nanofibers were electrosprayed with microparticles of various densities to provide topographic cues for cell contact and guidance. This nanofiber successfully directed the axon outgrowth of PC12 and SH-SY5Y neuronal cells, alignment of Schwann cells, and acceleration of neural stem cells migration when loaded with NGF in vitro [193]. A PLGA-electrospun nanofiber was functionalized with the sphingolipid ceramide N-deacylase-hydrolyzed monosialotetrahexosylganglioside (LysoGM1), a ganglioside of neuronal membranes, to form a PLCA-LysoGM1 scaffold. The scaffold enhanced neuronal cell growth, guided neurite extension, and facilitated the regeneration of injured neurons in a rat model of TBI. PLGA-LysoGM1 fibers supported the migration and infiltration of endogenous neurons into the site of injury, suggesting that this scaffold may be a promising therapeutic strategy for brain injury [145]. Cell-free biomimetic scaffolds are made of radially aligned electrospun poly-L/dL lactic acid (PLA70/30) nanofibers which can release L-lactate. PLA nanofiber architecture supports neuronal migration, and L-lactate released following degradation of the scaffold acts as an alternative energy source for neuronal progenitors. PLA radial scaffolds engrafted at brain cavities in a mouse model of TBI supported vascularization and neurogenesis, and allowed for the survival and integration of newly generated neurons. These results suggested that biophysical and metabolic signals support in vivo dedifferentiation of endogenous cells [149].

### 3.2. Mechanisms of Repair by Biomaterials in TBI

Functionalized biomaterials can offer a diverse array of therapeutic effects to address the multifaceted nature of TBI. These biomaterials can act through several mechanisms, such as promoting neurogenesis [146,194], enhancing the differentiation of implanted and endogenous neural stem cells into neurons rather than astrocytes or glia [165], inhibiting apoptosis [171,173], triggering anti-inflammatory effects and reducing neuroinflammation [195], curbing oxidative stress and exhibiting antioxidant properties [196], and facilitating angiogenesis [197].

Neuroinflammation in particular has a dual role post TBI. It can have both beneficial effects, such as facilitating cell debris removal, and detrimental effects, including the induction of neuronal death and neurodegeneration. Hence, anti-inflammatory therapy represents a viable approach for treating TBI [198]. Li et al. showed that employing an injectable enzymatically digested gelatin hydrogel loaded with mouse MSCs differentiated into the neuronal lineage significantly reduced the damaged area, mitigated inflammation, and lessened neuronal apoptosis in vivo in a mouse model of intermediate TBI [144]. Similarly, an injectable self-assembling nanofibrous peptide hydrogel effectively lowered acute brain injury by reducing the number of apoptotic cells, suppressing inflammation, and supporting cell survival [199]. Liu et al. demonstrated that the addition of HA to chitosan-based hydrogels (CH) reduced the expression of inflammatory markers and cell apoptosis, suggesting that the CH hydrogel effectively inhibited cell death at the injury site [171].

Hypoxia at the site of a traumatic brain injury can set off a chain of detrimental events, such as triggering glutamate excitotoxicity and the influx of Ca^2+^, which subsequently leads to the generation of free radicals and oxidative stress [200]. Consequently, the introduction of a biomaterial possessing antioxidant properties can be particularly advantageous. An injectable scaffold, HGA, created by combining HA–tyramine (HT) polymer with the antioxidant molecule gallic acid, significantly improved the neurological functions of mice with TBI. The effects of HGA were attributed to its ability to suppress oxidative stress through the activation of the Nrf2/HO-1 pathway [119]. Such biomaterials have the potential to counteract TBI-induced hypoxia by suppressing oxidative stress.

Biomaterials can enhance neurogenesis in TBI. Ma et al. created a hydrogel composed of sodium alginate, collagen, and stromal cell-derived factor-1 (SA/Col/SDF-1) and loaded it with human bone marrow MSCs (hBMSCs). The combination was able to enhance the recovery of neurological function after TBI by promoting neurogenesis in the hippocampus, indicated by an increase in EdU+/NeuN+ cells and the expression of neurotrophic factors [201]. Shi et al. introduced human-umbilical-cord-derived MSCs (hUC-MSCs) and activated astrocytes into a self-assembled peptide hydrogel scaffold called RADA16-BDNF (R-B-SPH scaffold). This scaffold stimulated the regeneration of neurons and the reconstruction of neural networks following TBI [202].

Angiogenesis can also be enhanced by biomaterials. RADA16-SVVYGLR is an SAP functionalized with the angiogenic motif SVVYGLR. When used in a zebrafish model of TBI, this SAP hydrogel was found to have programmable physical properties, biocompatibility, and regenerative properties that led to restoration of functional defects in TBI zebrafish [203]. Functional motifs were added to SLanc, an SAP hydrogel, to mimic vascular endothelial growth factor (VEGF-165). When tested in a rat model of TBI, SLanc induced more blood vessel formation than in the sham or TBI control rats. VEGF-receptor 2 was activated and protein levels of vascular markers, such as von-Willebrand factor (vWF) and α-smooth muscle actin, were elevated. The integrity of the BBB was restored, indicating that SLanc may offer the neuroprotection required for long-term recovery by inducing angiogenesis [163].

When used in combination with stem cells, biomaterials were shown to help implanted neuronal stem cells to survive post implantation by providing a microenvironment favorable for their growth. Neural stem cells and MSCs are the preferred cell source for neural regeneration in cell-based therapy of TBI. They can be combined with biomaterials and implanted directly at the site of injury [204]. Neural stem cells have the capacity of self-renewal and the ability to develop into neurons, astrocytes, and oligodendrocytes [204]. Therefore, they can replace the function of cells lost due to brain injury, making the transplantation of stem cells a promising way to restore the damaged CNS [179]. A suitable biomaterial offering a three-dimensional (3D) matrix or scaffold will be required to support the implantation and growth of transplanted cells and for the endogenous repair of damaged neural tissue; this is particularly crucial in brain tissue reconstruction. The biomaterial scaffold can fill in structural gaps, knit the injured brain back together, provide a substrate for neurite outgrowth, and act as a pathway for endogenous cells to migrate and axons to elongate [7]. Growth factors are needed to stimulate the neurogenesis of transplanted cells, and are usually loaded into the biomaterial [121]. In addition, the harsh microenvironment around the CNS injury site favors the differentiation of endogenous neural stem cells into astrocytes rather than neurons, contributing to the formation of the glial scar, which impedes recovery [41]. A conductive supramolecular hydrogel biomaterial was shown to provide an optimal niche for the neurogenesis of endogenous neural stem cells, thereby reducing glial scar formation and promoting repair of spinal cord injury [165]. Therefore, biomaterials and stem cells are also used to replace or augment the normal endogenous function of tissues, thus providing an efficient means for therapeutic intervention in TBI [121].

Collectively, many studies have indicated that combination treatments of cells, biomaterials, and bioactive molecules are more effective than treatments involving a single component. The combination of biomaterial-enriched microenvironments, neural stem cells, and growth factors can be used to increase functional recovery following TBI. These combination treatments are able to bypass the BBB in TBI, thus achieving local delivery to the brain and enhancing cell survival following cell transplantation. Wang et al. designed a hybrid hydrogel combining a self-assembling peptide and myoglobin protein, which effectively delivers oxygen to tissues. This hydrogel facilitated the delivery of both cortical neural stem cells and oxygen to mouse brain injury sites, supporting stem cell engraftment, survival, and integration for up to 28 days. The hybrid hydrogel outperformed a myoglobin-free version of the hydrogel in promoting engraftment, survival, and maturation of neuronal stem cells into neurons, while also encouraging blood vessel growth in the endogenous tissue. This study suggested that oxygen release from the hydrogel is linked to functional stem cell integration, offering a general strategy for developing improved tissue-mimicking biomaterials [205].

### 3.3. Complications, Limitations and Recommendations

In the field of CNS regeneration, developing biomaterials that are biodegradable, biocompatible, and mechanically flexible is still a major challenge. The design and use of these materials will require precise control over their delivery in order for them to exert a therapeutic function in the right place at the right time. Furthermore, creating materials that exhibit the same or similar delicate balance of structural and biological properties as the natural tissues is another hurdle. Other challenges, including the necessity to improve mechanical strength, durability, and stability during application, as well as the balance between fluidity and mechanical strength, limit biomaterial use. To add, creating biomaterials presents a fundamental technical challenge in preserving the free flow of nutrients, oxygen, and medicinal substances produced by enclosed cells, while preventing the swelling and eventual rupture of the cells [206].

Biomaterials from natural sources have flaws as well. Batch inconsistency is one of them, resulting in difficulties in controlling the homogeneity of the resultant scaffolds. Additionally, there are more risks related to cryo-preservatives, process additives, and other residues, as well as patient-specific reactions such as allergies to antibiotics or other drugs. Furthermore, the original natural sources may contain pathogens that cause immunological reactions [207]. Incompatible materials can cause unanticipated inflammatory responses against foreign bodies, leading to implanted tissue necrosis and/or rejection. Another complication is the degradation process. In vivo, synthetic polymers lack intrinsic biological properties, and their breakdown products may have negative effects or impact the local microenvironment [208]. If degradation products are created, they should be eliminated from the body through the metabolic path at a rate that keeps their concentration in the body at a manageable level [209]. Some biomaterials dissociate in a slow and unpredictable fashion [210]. The rate of biodegradation within an organism is determined by the polymer properties and the location in the body where it will be exposed. Composition and molecular structure, polydispersity, crystallinity, surface area, and hydrophilic or hydrophobic properties all impact chemical degradation. Chemical degradation affects major polymer chains by causing random breakage of covalent bonds, depolymerization or crosslinking of linear polymers, interference with regular order and crystallinity, and reduction in mechanical characteristics [211,212,213]. If the biomaterial remains in place for longer than intended, the residual material may inhibit rather than promote tissue regeneration. This suggests that biomaterial absorption kinetics will have a significant impact on tissue engineering success rates [130]. Moreover, while biomaterials are generally not considered to be carcinogenic, there are instances where their interaction with living tissue can trigger biological responses that may contribute to the development of cancer [214]. This highlights the need for further research on the long-term side effects of biomaterials. Table 3 provides a concise overview of the advantages and disadvantages associated with the use of biomaterials in tissue engineering.

The use of biomaterials is a novel approach in the treatment of TBI. As mentioned, despite the huge advances in formulating biocompatible, biodegradable, and mechanically flexible biopolymers, the available biomaterials still have limitations and the perfect biomaterial for use in human TBI patients has not been designed yet. Despite these limitations, studies based on several in vivo animal models of TBI have been published with encouraging results. However, there are no clinical trials of the use of biomaterials in TBI currently listed at clinicaltrials.gov. Most of the TBI clinical trials listed use a form of stem cell-based therapy alone without a biomaterial. In the case of stroke, clinical trials that used cell therapeutics alone without a biomaterial showed that cells directly implanted into the damaged brain do not survive due to the inflammatory environment of stroke, attacks by immune cells, and the absence of cues for cell adhesion and propagation [215]. Similar mechanisms may be present in the case of TBI. Other challenges for the use of biomaterials in human studies include immune- and hemo-compatibility, pyrogenicity, and certain undesirable properties of biomaterials, such as the lack of adhesion of some biomaterials to several cell types or degradation products of other biomaterials which may disturb the microenvironment of the host tissue [56,216,217]. Biomaterials have also been linked with substantial side effects as their implantation may require surgical operation, for example to remove neural tissues to accommodate a stiff scaffold. This may aggravate the existing injury [125,126,127,128,129]. In addition, tissue engineering techniques, particularly cell- and gene-based therapy, involve the manipulation of living cells and their interactions with substrates and biomolecules, potentially resulting in contamination and process failures. There may also be catastrophic repercussions of cell–substrate interactions that are yet to be discovered [218].

To address these limitations, several key solutions can be implemented [219,220,221]. These include the development of minimally invasive surgical techniques to reduce the invasiveness of and the complications associated with implanting the scaffolds. In addition, rigorous contamination controls can be applied to ensure uncomplicated recovery. Future efforts should focus on developing inherently biocompatible materials and controlling the biodegradation rates of biomaterials to ensure their stability during therapy to provide physical support for the injured tissue and sustain the extended release of therapeutic agents [210,212]. Biomaterials can be designed to closely mimic natural tissue properties. Multiple studies have demonstrated that this approach has led to enhanced biocompatibility and better outcomes [139,164]. In this regard, self-assembling peptide hydrogels exhibit promising opportunities to address several of these challenges. They have ECM-like biomimetic 3D structures, their degradation products are amino acids with good biocompatibility, and many of their properties can be tuned to cope with the hostile microenvironment of TBI [217]. Furthermore, biomaterials can contribute to the creation of an enriched environment by acting as a delivery agent for drugs, cells, as well as growth and neurotrophic factors. In cell-based therapeutics, biomaterials enhance cell retention and integration at the site of injury. Although this method holds promise in replacing lost neurons and rebuilding damaged neural pathways, it is not free of shortcomings. Challenges of biomaterial-assisted cell transplantation include limited survival due to the complex pathology and hostile microenvironment of TBI and the inefficiency of cell delivery and implantation in the damaged brain areas, mainly due to the presence of the BBB [222].

**Table 3 biology-13-00021-t003:** Advantages and challenges of the use of biomaterials in tissue engineering.

Biomaterial	Characteristic	Advantages	Disadvantages	References
Natural hydrogels	Cross-linked macromolecular networks	-No mechanical/spatial restrictions compared to synthetic polymer scaffolds -Mesh size and porosity of hydrogels can be modified -Biocompatible -Injectable -Porous	-Heterogeneity between batches -May carry natural pathogens -Difficulty in precise modification of the material	[119,146]
Synthetic hydrogels	Can be modified according to need	-Biologically inert -Chemically stable -Easier to control important perimeters	-Premade, require invasive implantation surgery -Cause more inflammatory response than natural hydrogels	[223,224,225]
Self-assembling peptides SAPNs	Composed of repeating units of amino acids and characterized by the formation of double-β-sheet structures	-High porosity -Increased cell signaling from bioactive peptides that are present in high density at the damaged site -Highly biocompatible -Allow minimally invasive treatments	-Lack of understanding of their degradability -Lack of data on long-term electroactivity of the scaffold	[182,226,227]
Electrospun nanofibers	A nonwoven mat of micro- and nanofibers is created when fluid filament is stretched in a powerful electric field	-Aligned nanofibers can resemble the topographical characteristics of the extracellular matrix in the brain -Due to large surface-to-volume ratio, electrospun fibers improve cell adhesion, mass transfer characteristics, and drug loading	-pH difference, local enzymes may degrade the fibers	[150,228]

## 4. Conclusions

Although TBI remains a leading cause of mortality and disability worldwide, it still lacks a definitive treatment. The use of biomaterials is emerging as a promising treatment approach to TBI. These innovative therapies involve the repair of damaged brain tissue using hydrogels, including self-assembling peptides, and electrospun nanofibers. These biomaterials can be used alone or in combination with stem cells, neurotrophic and growth factors, ECM proteins, antioxidants, oxygen delivery moieties, angiogenic factors, and other functional moieties. These biomaterials have shown great potential in reducing oxidative stress, neuroinflammation, apoptosis, the formation of astrocytes and glial scars, and rupture of the BBB, promoting neurite outgrowth, neural stem cell neurogenesis and differentiation, neovascularization, synapse formation, and nerve gap bridging in models of TBI. Whereas animal models have demonstrated the effectiveness of these tissue engineering strategies in treating TBI, there are still insufficient data on their efficacy in human subjects. Therefore, these biomaterials have not yet been used for treating TBI in a clinical setting. Challenges that persist the development of biodegradable, biocompatible, and mechanically adaptable biomaterials, and, in the case of a combination of biomaterials with cell-based therapeutics, the inefficient survival of transplanted neuronal stem cells in the harsh environment of TBI. Functionalization and the use of multi-functional supramolecular scaffolds of these biomaterials is steadily overcoming these limitations. As of now, biomaterials offer a valuable opportunity to explore new therapeutic avenues for TBI beyond traditional drug delivery and treatment methods.

## Figures and Tables

**Figure 1 biology-13-00021-f001:**
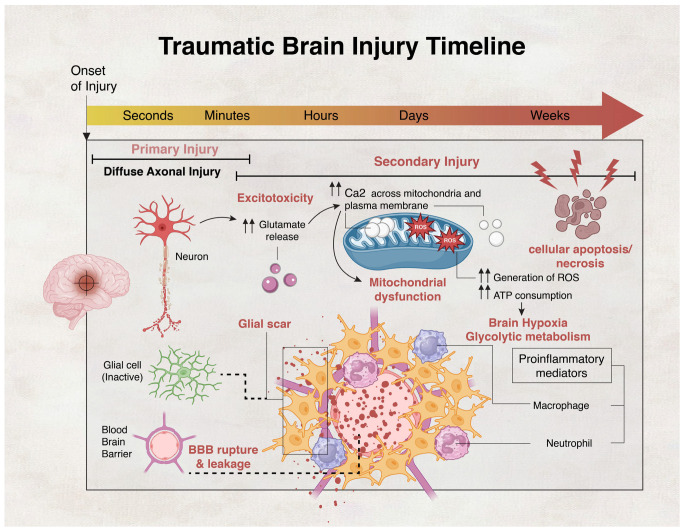
Timeline of the events of the primary and secondary phases of traumatic brain injury (TBI). Molecular events of the secondary phase start with an increase in the release of neurotransmitters, prominently glutamate, which leads to excitotoxicity, and Ca^2+^ influx into the cytoplasm and mitochondria of injured neurons. High Ca^2+^ levels can lead to mitochondrial dysfunction, including an increased consumption of ATP, excessive generation of reactive oxygen species (ROS), and cellular death by apoptosis and necrosis. The BBB ruptures and astrocytes numbers increase excessively leading to the formation of a glial scar that impedes neurotransmission. Neuroinflammation also takes place due to the activation of microglia, which secrete inflammatory mediators that serve to recruit immune cells from the periphery, such as neutrophils and macrophages. Mitochondrial dysfunction and the increase in ATP consumption can precipitate brain hypoxia, leading to a shift towards an inefficient glycolytic metabolism.

**Figure 2 biology-13-00021-f002:**
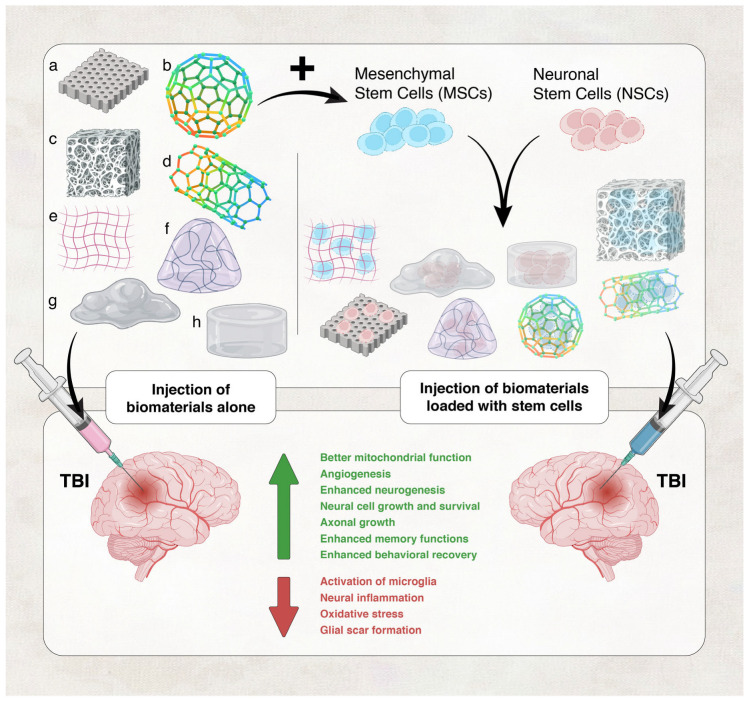
Use of biomaterials in TBI. These biomaterials can be used alone (**left panel**) or in combination with stem cells and other factors (**right panel**) to exert therapeutic effects in TBI. (**a**) Chitosan 3D-printed scaffold; (**b**,**d**) electrospun nanofiber scaffold; (**c**) macroporous scaffold; (**e**) engineered Chondroitin sulfate (eCS); (**f**) micro-ribbon hydrogel (soft); (**g**) hydrogel (soft); and (**h**) hydrogel (stiff).

**Figure 3 biology-13-00021-f003:**
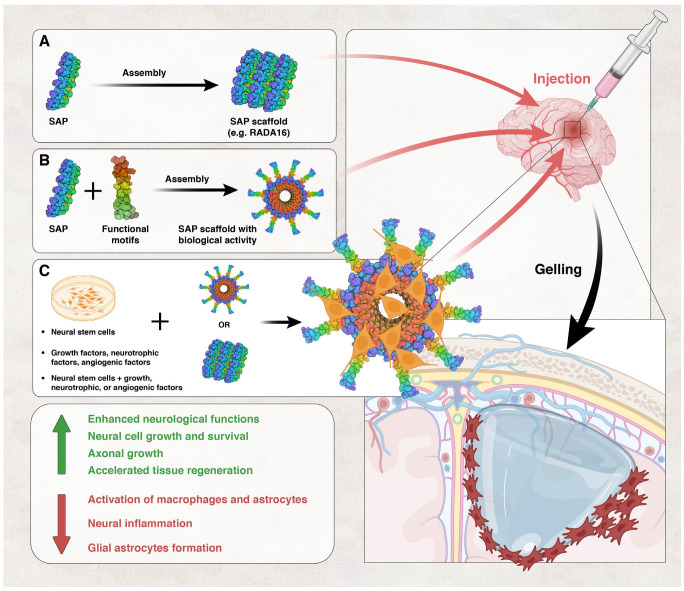
Use of self-assembling peptides (SAPs) in TBI. SAPs like RADA16 can be assembled into a scaffold that may be injected at the site of TBI (**A**). The SAP scaffold can gel at the site of injury, filling the cavities created by the injury. SAP scaffolds can also be assembled from SAPs containing functional peptide motifs, such as motifs with angiogenic or growth-promoting functions (**B**). SAPs or SAPs with functional motifs can also be assembled into scaffolds in the presence of stem cells and growth, angiogenic, neurotrophic, or ECM factors, or of combinations of stem cells and other factors (**C**). All these approaches have been used in animal models of TBI and were shown to have positive outcomes.

**Table 1 biology-13-00021-t001:** Main differences between biomaterials from natural or synthetic sources.

Property	Synthetic Biomaterial	Natural Biomaterial
Source	Artificially synthesized	Biological sources
Biodegradability	Variable, controllable	Naturally degradable
Immunogenicity	Generally low	Potential immune response
Mechanical properties	Customizable for specific needs	Variable
Biocompatibility	Reduced, can be optimized	Good biocompatibility
Growth factors	Controlled release	Potential endogenous release
Examples	Poly-anhydrides and poly-orthoesters.	Collagen, chitosan, hyaluronic acid

**Table 2 biology-13-00021-t002:** Studies testing the therapeutic potential of biomaterials for TBI.

Study	Biomaterial	Species	Outcome	References
Liu et al. (2023)	Collagen/chitosan/BMExos scaffold	Rat	Improved recovery of neuromotor function and cognitive function Facilitated remodeling of neural networks Improved regeneration of nerve fibers, synaptic connections, and myelin sheaths in lesions after TBI	[143]
Li et al. (2021)	Gelatin hydrogel	In vitro & mice	Reduced the damaged area Ameliorated inflammation Reduced neuronal apoptosis Enhanced survival and proliferation of endogenous neural cells Promoted functional recovery of motor, learning, and memory ability	[144]
Tang et al. (2020)	aPLGA-LysoGM1 scaffold	In vitro & rat	Promoted neuronal viability Enhanced neurite outgrowth Facilitated synapse formation Protected neurons from pressure-related injury	[145]
Zheng et al. (2020)	Gelatin methacrylate hydrogel with polydopamine nanoparticles and hAMSCs	Rat	Facilitated the regeneration of endogenous nerve cells Promoted differentiation of hAMSCs into nerve cells	[146]
Mahumane et al. (2020)	N-acetylcysteine (NAC)-loaded poly(lactic-co-glycolic acid) (PLGA) electrospun nanofiber	In vitro & ex vivo (Rat pheochromocytoma PC12 cells) and human glioblastoma multiform A172 cells)	Improved cell viability and proliferation Found to be neuroprotectant in a spatial and temporal manner	[147]
Zhou et al. (2018)	poly(lactic-co-glycolic acid) (PLGA) scaffold	In vitro & in vivo Mesenchymal stem cells (MSCs) and neurons	Better cell adhesion on the MSC-PLGA scaffold complex than the PLGA scaffold MSCs could migrate out to the adjacent brain area	[148]
Álvarez et al. (2014)	poly-L/DL lactic acid (PLA70/30) nanofibers	Mice	Induced the generation of several types of neurons and glial cells and established synaptic contacts New neurons survived for more than a year	[149]
Sulejczak et al. (2014)	Electrospun nanofiber/L-lactide-caprolactone copolymer nanofiber net	Rat	Prevented astrogliosis and inflammation initiation Resulted in a thinner, more orderly scar formation Produced lactic acid, a neuroprotective factor	[150]
Zhang et al. (2018)	Vepoloxamer	Rat	Reduced cortical lesion volume by 20% Activated microglia/macrophages and astrogliosis in different brain regions Normalized bleeding time and reduced brain hemorrhage and microthrombosis formation	[151]
Macks et al. (2022)	poly(Ethylene) glycol-bis-(acryloyloxy acetate) (PEG-bis-AA) with dexamethasone (DX)-conjugated hyaluronic acid (HA-DXM)	Rat	Improved motor function recovery at 7 days post injury as assessed by the rotarod test Enhanced cognitive functional recovery as assessed by the Morris water maze test Reduced apoptosis and lesion volume compared to untreated animals at 14 days post injury	[152]
Latchoumane et al. (2021)	Engineered Chondroitin sulfate (eCS)	Rat	Significantly enhanced cellular repair and gross motor function recovery at 20 weeks post injury Noticeable recovery of “reach-to-grasp” function	[134]
Liu et al. (2022)	Secretome/collagen/heparan sulfate scaffold	Rat	Improved cognitive and locomotor function as demonstrated by modified Neurological Severity Score (mNSS), Morris water maze (MWM), and motor-evoked potential (MEP) tests Enhanced reconstruction of neural structures and reduced apoptotic response and neuroinflammation at the injury site	[153]
Sahab Negah et al. (2019)	Self-assembling peptide hMgSCs + R-GSIK	Rat	Increased number of hMgSCs in the brain compared to control groups Reduced lesion volume, reactive gliosis, and apoptosis at the injury site Suppressed expression of Toll-like receptor 4, interleukin-1β, and tumor necrosis factor	[154]
Liu et al. (2023)	Bone marrow mesenchymal stem cell-derived exosomes (BME) + hyaluronan-collagen hydrogel (DHC-BME)	Rat	Promoted endogenous NSC recruitment and neuronal differentiation Promoted axonal regeneration, remyelination, synapse formation, and brain structural remodeling Inhibited astrocyte differentiation Induced angiogenesis and neurogenesis	[155]
Tanikawa et al. (2023)	Electrically charged hydrogels (C1A1) + VEGF	Mice	NSCs differentiated into neuroglial cells Promoted formation of host-derived vascular network Infiltration of macrophages/microglia and astrocytes into the C1A1 hydrogels was observed	[156]
Hu et al. (2023)	Self-healing hydrogel (HA-PBA/Gel-Dopa)	Mice	Supported neural cell infiltration Decreased astrogliosis and glial scars Induced lesion sealing	[157]
Moisenovich et al. (2019)	Silk fibroin scaffold	Rat	Scaffold demonstrated neuroprotection by promoting recovery of neurological functions and restoration of sensorimotor functions Reduced the volume of damage caused by traumatic brain injury (TBI) by 30% compared to the control group Fibroin-gelatin microparticles restored neurological status by 25% starting from the 4th day after TBI induction	[158]
Chen et al. (2022)	Hydrogen sulfide(H_2_S)-releasing silk fibroin (SF) hydrogel (H_2_S@SF)	Mice	Significantly reduced TBI-induced neuronal pyroptosis Inhibited the expression of the necroptosis protein Alleviated brain edema and neurodegeneration in the acute phase of TBI	[159]
Jiang et al. (2021)	Collagen/Silk fibroin (SF) scaffold	Canine	Improved cerebral cortex integrity and motor functions Regeneration of blood vessels and nerve fiber Reduced glial fibers and inflammatory factors	[160]
Qian et al. (2021)	TM/PC hydrogel (tri-glycerol monostearate, propylene sulfide, and curcumin)	Mice	Hydrogel demonstrated ROS scavenging ability and protected BBB integrity Reduced brain edema Promoted M1 macrophage polarization	[161]
Zhang et al. (2022)	HT/HGA hydrogel (hyaluronic acid-tyramine + antioxidant gallic acid-grafted hyaluronic acid)	Mice	Suppressed expression of proinflammatory factors and promoted the polarization of microglia from M1 to M2 phenotypes Protected cells from oxidative stress induced by H_2_O_2_, reducing intracellular ROS production and promoting cell viability	[119]
Chen et al. (2023)	Gelatin methacrylate and sodium alginate hydrogel (GelMA/Alg)	Rat	Reduced tissue loss and alleviated astrogliosis Significantly decreased microglial activation and neuronal death in the acute phase of traumatic brain injury (TBI) Improved neurological recovery in the chronic phase of TBI	[162]
Ma et al. (2020)	Self-assembling peptide-based hydrogel	Rat	Significantly more blood vessels were observed in the SAPH-treated group Immunohistochemistry and biomarkers confirm angiogenesis and activation of VEGF-R2 Neuroprotection and axonal outgrowth were observed in and surrounding the hydrogel	[163]

## Data Availability

Not applicable.

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
