# Peer review of "Biomaterials in Traumatic Brain Injury: Perspectives and Challenges"

_biology, 2023, doi:10.3390/biology13010021_

Round 1

Reviewer 1 Report

 The manuscript by Shaito et al summarizes the current literature on the therapeutic strategies of biomaterials for traumatic brain injury. Overall, this manuscript covers relevant, updated, and adequate references.  However, server major concerns must be addressed before this manuscript can be published.

1.    The scope of this manuscript is vague. The authors mentioned biomaterials and cell delivery multiple times in the title, abstract, introduction, and section 2.1. However, the overall scope of this manuscript is not clarified. And Section 2.2-2.4 are mainly focused on biomaterials aspects. Which topic is focused on, all biomaterials, biomaterials for cell delivery, or both biomaterials and cell therapeutics? The authors need to specify the scope and keep focusing on the topic.

2.    The structure of section 2 is confusing. The manuscript should be organized and classified based on consistent characteristics. For example, if this manuscript focuses on biomaterials, the organization may include chemical properties (e.g. synthetic or native materials), physical properties (e.g. dimension, porosity, alignment), and biological properties (e.g. cell attachment, cell fate direction, immunomodulation).  If this manuscript is focused on biomaterials and cell therapeutics, the organization can be scaffold and cell types.

However, this manuscript

(a)  all properties are mixed up: Section 2.1 mixed with partial discussions about pathways, materials, and cells. Section 2.2 mentioned biomaterials. Section 2.3 also discussed the materials.

(b)  The sections are classified based on variable characteristics: Section 2.3.1 described hydrogel based on hydrophilicity. Section 2.3.2 described self-assembling peptides based on the material type. However self-assembling peptides can form hydrogels. Section 2.3.3 described nanofibers based on dimensions. The materials mentioned in Section 2.3.1 and 2.3.2 may also be used for nanostructures.

In this context, Section 2 should be restructured.

3.    More discussion, including critical issues that remain in the field, how to solve the issues, future directions, etc., should be included. 

Minor issues:

1. Biomaterials demonstrated in Figure 1 need to be labeled in the figure or specified in the figure legend.

2. Line 137, Line 156: The sub-titles are confusing since biomaterials serve as an approach for tissue engineering purposes.

3. Formatting of the text should be consistent. Examples included in line 278, Line 286-290, Line 308-210, Line 319-325.

The language of this manuscript is acceptable and easy to understand.

Author Response

* We are grateful for the thoughtful evaluation of our manuscript by both Reviewers. We also appreciate the time and effort spent by the editors and the Reviewers to review our manuscript. Suggestions by both Reviewers have enhanced the quality of the manuscript. Below is a point-by-point response to the Reviewers’ comments.

The manuscript by Shaito et al summarizes the current literature on the therapeutic strategies of biomaterials for traumatic brain injury. Overall, this manuscript covers relevant, updated, and adequate references. However, server major concerns must be addressed before this manuscript can be published.

* We would like to thank the Reviewer for his comments and suggestions and hope that our modifications meet with his approval.

  1. The scope of this manuscript is vague. The authors mentioned biomaterials and cell delivery multiple times in the title, abstract, introduction, and section 2.1. However, the overall scope of this manuscript is not clarified. And Section 2.2-2.4 are mainly focused on biomaterials aspects. Which topic is focused on, all biomaterials, biomaterials for cell delivery, or both biomaterials and cell therapeutics? The authors need to specify the scope and keep focusing on the topic.

* We thank the Reviewer for his comment. Following the Reviewer’s notice, we have now re-organized the manuscript to better align with the aims.

The aim of our manuscript is to provide a comprehensive review of biomaterials that showed positive outcomes when used in the context of TBI therapy.

We had to introduce ALL biomaterials first, then focus on the biomaterials that had “success” in TBI, either in cell culture, animal models. Upon the reviewers suggestions We have now changed section 2 to include general properties of biomaterials used in neurological diseases. Section 3 now includes the use of biomaterials in TBI only. We discuss all applications of biomaterials in TBI: 1) biomaterials alone or 2) in combination with cell-based therapeutics. It may seem that we have focused more on biomaterials combined with cell-based therapeutics in TBI therapy, but is due to the fact that there are more studies on the use of biomaterials in combination with cell-based therapeutics, than biomaterials alone.

  1. The structure of section 2 is confusing. The manuscript should be organized and classified based on consistent characteristics. For example, if this manuscript focuses on biomaterials, the organization may include chemical properties (e.g. synthetic or native materials), physical properties (e.g. dimension, porosity, alignment), and biological properties (e.g. cell attachment, cell fate direction, immunomodulation). If this manuscript is focused on biomaterials and cell therapeutics, the organization can be scaffold and cell types.

However, this manuscript

  • all properties are mixed up: Section 2.1 mixed with partial discussions about pathways, materials, and cells. Section 2.2 mentioned biomaterials. Section 2.3 also discussed the materials.

*  We thank the Reviewer for his valuable comment, we have re-organized the manuscript as mentioned in the previous comment. All physical and chemical properties of biomaterials are now in section 2, the use of biomaterials in neurological disorders. Specific applications of biomaterials in TBI are now present in section 3.

(b)  The sections are classified based on variable characteristics: Section 2.3.1 described hydrogel based on hydrophilicity. Section 2.3.2 described self-assembling peptides based on the material type. However self-assembling peptides can form hydrogels. Section 2.3.3 described nanofibers based on dimensions. The materials mentioned in Section 2.3.1 and 2.3.2 may also be used for nanostructures.

In this context, Section 2 should be restructured.

*  The manuscript has been restructured as suggested. In addition, self-assembling peptides are now included under hydrogels. We thank the reviewer for noticing this.

  1. More discussion, including critical issues that remain in the field, how to solve the issues, future directions, etc., should be included.

* We thank the reviewer for this comment. Accordingly, we have changed the subtitle of section 3.3 to better reflect the content. Section 2.3 now discusses the limitations of using biomaterials in general and in TBI and the possible solutions to these challenges. We have also discussed the absence of TBI clinical studies including biomaterials.

Minor issues:

  1. Biomaterials demonstrated in Figure 1 need to be labeled in the figure or specified in the figure legend.

* The biomaterials have been labelled in the figure as a, b, c and then indicated in the figure legend. Thank you for this suggestion.

  1. Line 137, Line 156: The sub-titles are confusing since biomaterials serve as an approach for tissue engineering purposes.

* We have amended the subtitles.

  1. Formatting of the text should be consistent. Examples included in line 278, Line 286-290, Line 308-210, Line 319-325.

* We have unified the text font. Thank you for pointing this out.

We appreciate the time, thoughts, and effort spent by the Reviewer on this evaluation and hope that our modifications meet with his approval.

Reviewer 2 Report

The authors have described the different types of nanomaterials used for the delivery of drugs, stem cells etc. in the injured brain to revert the symptoms of TBI.

The manuscript is well written and the recent developments in the management of TBI is elaborately discussed. The disadvantages of biomaterials to be used in TBI is also discussed.  I have a few minor comments:

The entire manuscript has only one figure. Please include at least two more high quality figures for better understanding on the topic. One of the figures can be representing the role of self assembled peptides in reverting the brain injury.

The authors are suggested to discuss the relation between self assembled peptides for recovery of brain injury and the beta sheet rich amyloid fibrils of Aβ 1-42 which are also self assembled but pay a vital role in the onset of Alzheimer’s disease.

Minor comments:

The font is different in different place inside the text. Please unify them.

I recommend minor revision.

In one or two places small corrections are necessary. Else, the language is very good.

Author Response

* We are grateful for the thoughtful evaluation of our manuscript by both Reviewers. We also appreciate the time and effort spent by the editors and the Reviewers to review our manuscript. Suggestions by both Reviewers have enhanced the quality of the manuscript. Below is a point-by-point response to the Reviewers’ comments.

The authors have described the different types of nanomaterials used for the delivery of drugs, stem cells etc. in the injured brain to revert the symptoms of TBI.

The manuscript is well written and the recent developments in the management of TBI is elaborately discussed. The disadvantages of biomaterials to be used in TBI is also discussed. 

*We would like to thank the Reviewer for his comments and suggestions. 

I have a few minor comments:

  1. The entire manuscript has only one figure. Please include at least two more high quality figures for better understanding on the topic. One of the figures can be representing the role of self assembled peptides in reverting the brain injury.

*A figure of the molecular events during TBI and another figure on how self-assembled peptides can help revert TBI have been included in the manuscript. The figure on self-assembled peptides may have similarities with the figure on biomaterials since they follow similar mechanisms. But, we tried our best to distinguish them. We thank the Reviewer for his recommendation.

The authors are suggested to discuss the relation between self assembled peptides for recovery of brain injury and the beta sheet rich amyloid fibrils of Aβ 1-42 which are also self assembled but pay a vital role in the onset of Alzheimer’s disease.

*  We thank the reviewer for his valuable suggestion. It is an interesting research idea as well. Alzheimer’s disease is one of the long-term consequences of TBI, but we could not make comparison between Aβ 1-42  and SAPS in relation to TBI. Therefore, we did not include it in the discussion.

Minor comments:

The font is different in different place inside the text. Please unify them.

*  We have unified the text font. Thanks for pointing this out.

I recommend minor revision.

*  We appreciate this evaluation by the reviewer and hope that our modifications meet with his approval.

We greatly appreciate your kind consideration.

Sincerely,

Round 2

Reviewer 1 Report

All issues are addressed.